# Research culture in biomedicine: what we learned, and what we would like to do about it

Alexa T. McCray, David Van Vactor, James Gould, Xiuqi Li, Jelena Patrnogić,
Caroline Shamu & Mary C. Walsh

The significant stressors that affect the biomedical research community have the potential to negatively impact the conduct of science. Here we report on work done at Harvard Medical School to identify areas for improvement in research rigor, reproducibility, and responsibility in pursuit of continued research excellence.

Over the last several years, we have worked to better understand the research culture at a large, complex medical school dedicated to research and training. While Harvard Medical School (HMS) is a leading institution for biomedical research and training, like many of our peer institutions in this rapidly evolving area of science, its academic and research community is subject to a variety of stressors and challenges. Some of these factors have the potential to negatively affect research conduct and productivity, laboratory culture, individual satisfaction or advancement opportunities, and, importantly, the reliability of research findings and public trust in science itself[1,2]. The many demands on laboratory leaders in today's hypercompetitive research environment can contribute to inadequate mentorship practices and suboptimal laboratory oversight. In addition, there continues to be a need for greater national consensus on best practices for training in principles and practices of rigorous research. Notably, the traditional incentive structure of academic advancement often fails to adequately reinforce practices that promote rigor, reproducibility, and responsibility (R3) in research[3]. Thus, the forces that shape biomedical science careers and the reliability of scientific discoveries are intimately connected.

Recent surveys, workshops, and other studies have revealed concerns about the state of scientific research culture. Findings indicate that laboratory environments that are supportive, productive, collaborative, and rigorous are not only good for laboratory members, but also lead to better science[4–8]. Conversely, an unhealthy laboratory environment can lead to questionable research practices, or, in some cases, to researchers leaving science entirely[9–12]. The need for better career development support and incentive structures have become acute issues for trainees faced with the difficult choice of remaining in academia or pursuing careers in the private sector. The Director of the Wellcome Trust, after reviewing the results of their research culture survey involving thousands of researchers in the UK, commented that "poor research culture ultimately leads to poor research"[13]. Greater transparency in science has been promoted as one countermeasure, with institutions being encouraged—through consensus studies, guidelines, mandates, and grassroots efforts—to create a research culture that better supports and rewards researchers who are engaged in transparent research practices[14–21].

Because effective mentorship is a key aspect of a healthy research culture, methods for improving and strengthening approaches to mentorship have been recommended[22–24]. One of the critical characteristics of strong research leaders, or "research exemplars," as they have been called, is their excellence in mentoring[25]. Guidelines for the appropriate treatment of research trainees have been developed by the Association of American Medical Colleges[26–28]. The Center for the Improvement of Mentored Experiences in Research uses a train-the-trainer model for mentors and mentees at all career stages throughout the US, and the related National Institutes of Health (NIH)-supported National Research Mentoring Network provides a host of mentorship resources with a focus on diversity and inclusivity[29,30].

Although instruction in the principles and practices of rigor, reproducibility, and responsibility (R3) in research has been a component of traditional training programs in biomedical science for many decades, the methods and modalities of training have received renewed attention at academic institutions that rely on federal funding[31–36]. Several NIH institutes have issued funding opportunity announcements to develop, pilot, and disseminate training modules to enhance scientific rigor and data reproducibility, not only for training grant programs, but also for investigator-initiated grant programs[37–39]. The current trend towards a more data-driven approach to optimizing the design and delivery of science training provides a rational basis to assess efficacy, and a potential means to improve transparency and accountability for program outcomes. However, national funding agencies provide relatively little by way of structured frameworks and specific guidelines for training, perhaps to allow institutions more freedom to develop innovative approaches. Moreover, external sources of funding to propel and sustain new programs of training beyond individual support for trainees are largely lacking, thus forcing institutions of higher education to be resourceful and creative in developing solutions. All of this suggests that a vigorous effort is required within each institution, and that sharing of new ideas and approaches across peer institutions can provide an economy of innovation as well.

Beginning in early 2019, with the support of our dean, we engaged our colleagues at HMS and its affiliated hospitals in a series of conversations and workshops about R3 in research. (See Figs. 1 and 2 for additional details.) We established an R3 working group that held initial meetings, and we developed a variety of R3 resources. Following a kick-off meeting in late 2019, a survey of the R3 working group and faculty advisory committee asked respondents why they were interested in being involved in the R3 effort. Many felt that culture change in the practice of science was needed and that more attention should be paid to education and training in R3. Some expressed a desire for consistent R3 guidelines and policies. Additional comments spoke to HMS's values[40] and to the stressors that are in danger of eroding those values, and other comments related to providing

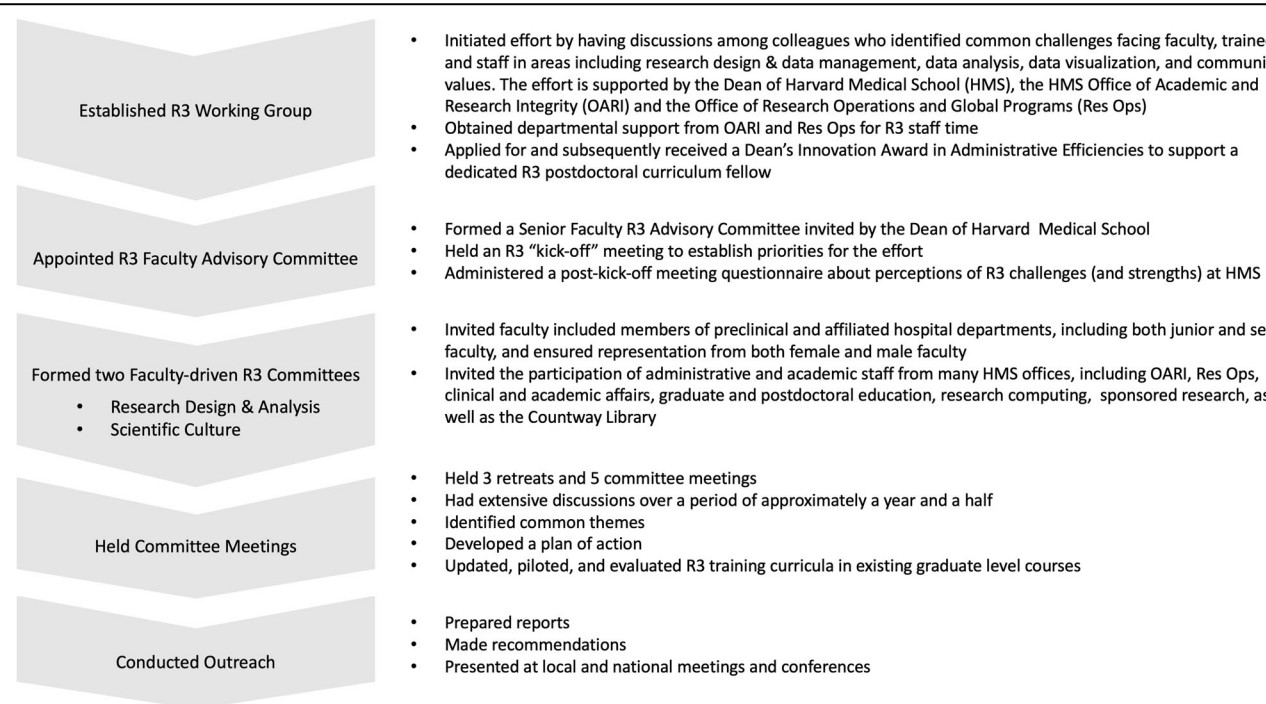

**Fig. 1 | An overview of the Harvard Medical School rigor, reproducibility, and responsibility (R3) in research effort (2019–2023).** The rigor, reproducibility, and responsibility in research effort has involved many members of the Harvard Medical School community and has proceeded with strong support from its leadership.

**Retreat and Committee Meeting Topics**

**Research Design & Analysis Retreat Topics**
Short talks on a variety of topics, e.g.,
- Embedding experimental design in a core course
- Teaching experimental design and analytics through technology platforms
- Curriculum development
- R3 skills and competency assessments
- Quantitative & computational resources
- Electronic Lab Notebooks

Discussion & brainstorming - What is missing and what future shared resources would have benefits across programs?

**Scientific Culture Committee Topics**
Discussions on a variety of topics, e.g.,
- Modeling and promoting a positive laboratory culture
  - How can we ensure healthy and productive lab environments?
  - What are the advantages and disadvantages of the typical mentoring model?
  - What is the best way to ensure that lab members recognize what situations and actions may result in questionable research practices?
  - Is R3/RCR training the optimal method for ensuring that all members of the lab conduct themselves in accordance with accepted professional conduct standards?
  - What resources are needed to ensure that data and other research results are captured and preserved for possible future review and use?
- Fostering high standards of professional conduct at the institutional level
  - What steps might be taken at the institutional level to address R3 issues such as those seen on faculty conduct committees?
  - Do you know of successful R3 approaches that other institutions or groups have taken that might be adopted/adapted?
  - Are there ways to help ease the ever more demanding requirements of the Federal government and other funders?
- Rewards, recognition, and metrics for rigorous and transparent research practice
  - Are there ways that academic advancement criteria could be enhanced to reward and incentivize best practices in research?
  - Can we facilitate ways for our researchers to recognize and engage in R3 endeavors as an element of successful career development and advancement?
  - What is the best way to measure whether R3 interventions have been successful?

**Fig. 2 | Sample topics and discussions at Harvard Medical School faculty-driven research rigor, reproducibility, and responsibility retreats and committee meetings.** Beginning in early 2019, colleagues at Harvard Medical School and its affiliated hospitals engaged in a series of conversations and workshops about rigor, reproducibility, and responsibility in research.

better support for mentors so that they, in turn, can best support their mentees.

We subsequently established two faculty-led committees and convened multiple meetings and retreats that included representation from HMS-wide academic departments. Participants included both junior and senior faculty (with a somewhat higher proportion of the latter), and an almost equal number of men and women faculty, as well as interested HMS administrative and academic staff, including representation from many HMS units and offices (e.g., Countway Library, Curriculum Fellows Program, Faculty Affairs, Graduate Education, Office for Academic and Research Integrity, Office for Postdoctoral Fellows, Research Operations, Research Cores and Technology, etc.).

We addressed a wide range of topics, including modeling and promoting a positive laboratory culture; developing best practices, resources, and skills in experimental design and data analysis; reinforcing research rigor in graduate and postdoctoral training; fostering high standards for professional conduct at the institutional level; and assessing and rewarding rigorous and transparent research practices. Our internal deliberations were informed by the extensive efforts to address R3 that are ongoing at the national and global level.

## What we learned

Over the course of our meetings and retreats, we had wide-ranging and frank conversations about today's biomedical research culture, and, in particular, what we might do in our own environment to effect positive change. Several major themes emerged as part of these discussions: (1) laboratory culture, (2) training and mentorship, (3) questionable research practices, (4) transparency of research results, and (5) institutional support and recognition.

**Laboratory culture**. Many of our conversations focused on the current culture in our pre-clinical, clinical and translational research laboratories. We recognized that the key to supporting a positive and healthy lab culture is to build and maintain a culture of respect in the lab. This includes welcoming diversity and supporting open communication among team members, and working to strengthen the connections between our work and the ongoing diversity, equity and inclusion initiatives in our community. We noted that it is also important to set expectations for research conduct in the lab. Some lab leaders have found laboratory manuals to be helpful. These provide important and continuously updated information about standard operating procedures, safety procedures, lab authorship policies, and contacts and resources that are available to lab members, to name a few.

We discussed other aspects of lab culture that may be undervalued, or perhaps even overlooked. We recommended the reinvigoration of social connections among colleagues, especially given the disruptions of the coronavirus disease 2019 pandemic. We also felt that it was important to create opportunities for non-work-related activities and to engage regularly in discussions to better assess laboratory health. Limiting email engagement during non-work hours, recognizing and supporting family commitments, and encouraging paid time off, can help contribute to the well-being of lab members. We discussed mental health issues at some length, including methods to promote mental wellness and normalize access to mental health services. One suggestion was to train and appoint mental health liaisons at the lab, program, or departmental level.

**Training and mentorship**. We posed the question of whether responsible conduct of research (RCR) training is the optimal method for ensuring that all members of the lab conduct themselves in accordance with accepted

professional conduct standards. There was general agreement that while such formal training is important, we need to be better at bridging the gap from the classroom to what actually happens in the day-to-day lab setting. Some of our faculty may not be fully aware of the curriculum that exists for students and postdoctoral scholars who enter their labs, and incentives for effectively teaching research design and analysis principles are often missing. We discussed the need to create a structure to organize and document RCR training that adapts well to different programs and to different career stages. Due to the tremendous diversity and scale of faculty research and training backgrounds, consistent transfer of best practices taught in our classrooms to everyday laboratory activities remains a challenging part of our training pipeline.

Given the importance of mentorship and the impact it can have on the career persistence and success of trainees, we discussed a variety of ways to strengthen approaches to mentorship, including considering and evaluating different mentorship models and providing ongoing training in mentorship best practices. Some suggestions included implementing a multi-mentorship model, where each mentor is available throughout the course of the training period and each mentor represents a different, but relevant area of expertise. Providing faculty with ongoing training in rapidly evolving technological advances in biomedicine was seen as critically important. There was strong support for the continuation and expansion of our existing formal faculty mentorship training program which provides faculty with strategies for effective guidance of their trainees as they progress through their decision making processes and career journeys.

**Questionable research practices**. We spent some time discussing how we can best help lab members recognize what situations and actions might result in questionable research practices before they occur[41–43]. The importance of focusing our efforts on preventing fires, rather than putting them out after the fact was underscored. Destigmatizing errors, setbacks, challenges, and failures by allowing protected spaces to discuss these can help avoid poor research practices. Regularly conducting lab figure review sessions during manuscript preparation, checking raw data, validating reagents, and recognizing where corners have been cut are all practices that may reveal questionable research practices that, if ignored, have the potential to become cases of research misconduct.

Illustrating questionable research practices through (anonymized) case studies in our training courses would help trainees see actual examples of these issues in our community. Creating a cadre of rigor champions[44] within labs and departments and recognizing rigor champion as a defined academic role could help ensure sustainability. We also gave some consideration to the idea of downsizing/right-sizing research labs so that mentor:trainee ratios are optimal for effective mentoring.

**Transparency of research results**. We discussed the importance of preparation and management of data to enable future reproducibility and sharing those data and materials openly, so that they are of maximal use to other investigators. There was discussion about the value of reporting on both positive and negative findings. We noted that transparency is enhanced by registering research protocols, by depositing preliminary results in openly available preprint archives (e.g., bioRxiv or medRxiv), by adhering to established funder and publisher data sharing and reporting guidelines and requirements, and by depositing data and publications in public repositories. We discussed exploring a variety of ongoing reproducibility efforts, such as the *eLife* initiatives for assessing experimental outcomes prior to publication.

Integrating conversations about R3 across our community will allow us to learn from our combined experiences, stressing that it is important to

balance and incentivize community engagement around priority areas, while being mindful of the potential burden of too many mandates.

**Institutional support and recognition.** We discussed the feasibility of ongoing institutional support for research data management, including support for wider adoption of electronic laboratory notebooks, laboratory information management systems, research protocol repositories, and templates for data management plans. Better tools for archiving data, tracking experimental metadata, and training in best data management practices could help decrease the management burdens of active labs, and help set consistent standards and operating expectations across our community. We endorsed the idea of providing institutional support for the hiring and training of professional data stewards. Data stewards would have discipline-specific expertise and would provide full-time data management support at the laboratory or departmental level. We also recommended that additional institutional as well as federal resources and support for implementation of new federal data management and sharing policies[45] be made available, including having individuals on-call to liaise with grant managers and to point researchers in the right direction, allowing troubleshooting without being held to sign-off.

We spent some time discussing the value of assessing and rewarding rigorous and transparent research practices as part of promotion and tenure decisions. De-emphasizing the role of the publisher-driven impact factor and articulating the multiple aspects of impact would help establish a system that rewards those aspects. We discussed the possible restructuring of the pre-tenure/pre-professorial reward system, with concomitant changes to the format of our curriculum vitae, such that data sharing, creating large-scale openly available datasets, protocol sharing, serving on local and national ad hoc committees and panels on research integrity matters, and other contributions to open science are considered as part of promotion and appointment.

## What we would like to do about it

We propose a multi-pronged approach to promoting R3 principles and practices across our own research community. We suggest four areas of action: (1) engage the community in a sustained and highly visible effort to promote rigor, reproducibility and responsibility in research; (2) provide an innovative data-driven training curriculum in R3 principles and practices; (3) support a positive and inclusive laboratory culture at HMS; and (4) recognize and reward R3 research practices. We are currently in discussions about what resources would be needed in order to make progress in our suggested areas of action.

**Engage the HMS-wide community in a sustained effort to promote R3 in research.** We envision an academic hub that supports evidence-based R3 research and training and provides innovative and impactful resources for students, trainees, faculty, and staff. We expect to collaborate with our many partners within HMS and affiliated institutions—and beyond—to support a sustainable, effective R3 effort. A critical component of any scholarship toward this goal must include continuous evaluation of the effectiveness and impact of R3 efforts, by developing and using a variety of metrics.

**Provide an innovative data-driven training curriculum in R3 principles and practices.** Although HMS has a long tradition of instilling in trainees the principles and practices of rigorous experimentation, rapidly advancing knowledge and technology demand that our training curriculum constantly evolve to meet exciting new opportunities and challenges at the vanguard of research. With our colleagues at HMS, we will look for opportunities to share best practices for R3 training and continue to build on the core competencies that have been defined by federally-funded training programs. We are currently designing a curricular framework that maps skills across diverse program offerings to competencies at distinct career stages, and we will develop assessment tools and instruments to allow for the improved collection and analysis of training data. To ensure a consistent transfer of theory to practice, we will work to bridge the gap between R3 training in the classroom and everyday laboratory practices by providing helpful materials for laboratory heads to use.

**Support a positive and inclusive laboratory culture at HMS.** Our community values include recognizing that our behavior affects the experiences of others, and that we value the well-being of every member of the community. Fostering a positive research laboratory culture, providing guidance and training for optimal mentorship relationships, and promoting scientific integrity in the research setting are paramount for excellence in science. To this end, we plan to examine and strengthen approaches to mentorship and work to model and promote a positive, diverse, inclusive, and equitable laboratory environment.

**Engage in and reward R3 research practices.** Progress in science depends on rigorous and transparent research practices. The ability to reproduce the results of a study allows others to assess the accuracy and trustworthiness of the results. While being mindful of excessive mandates, we will continue to work to maintain awareness of R3 principles and practices at all career levels, to improve laboratory procedures to prevent questionable research practices, and to promote best practices in data management and analysis. We hope to prompt a review of the academic reward system at HMS so that it might be updated to give appropriate credit to researchers who engage in rigorous and transparent research at all stages of research, from study and experimental design to reporting.

## Conclusion

Research rigor has always been central to the effective progress of science. However, the escalating speed of information gathering and dissemination has amplified the impact of errors that might previously have been corrected through the natural process of continuing inquiry. This acceleration is further exacerbated by a hypercompetitive system of incentives in academia and science publishing that emphasizes short-term advances and novelty, rather than the robust and circumspect advance of knowledge. Interestingly, at the same time, private sector investment in science and technology is at an all-time high, making industry increasingly appealing to science trainees. Therefore, in the current research climate, it is not only essential for academic institutions like ours to take a careful look at the way that we conduct and teach science to produce the next generation of reliable investigators but also to think deeply about what will sustain the future of academic research in the coming decades.

Alexa T. McCray [1,6] ✉, David Van Vactor [2,6], James Gould[3], Xiuqi Li [2], Jelena Patrnogić [2], Caroline Shamu [4] & Mary C. Walsh[5]
[1]Department of Medicine, Harvard Medical School, Boston, MA, USA. [2]Blavatnik Institute of Cell Biology, Harvard Medical School, Boston, MA, USA. [3]Office for Postdoctoral Fellows, Harvard Medical School, Boston, MA, USA. [4]Department of Radiology, Harvard Medical School, Boston, MA, USA. [5]Maidstone Consulting Group, LLC, Boston, MA, USA. [6]These authors contributed equally: Alexa T. McCray, David Van Vactor. ✉e-mail: amccray@bidmc.harvard.edu

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

## Acknowledgements
We thank George Daley, the Dean of Harvard Medical School, for initiating and supporting this effort. We are most grateful to the many faculty and staff who shared their insights with us. We thank our dedicated colleagues, Rachel Fouché, Julie Goldman, Jason Heustis, Melissa Korf, Jessica Pierce, and Daniel Wainstock, and we are indebted to Deans Kristin Bittinger and David Golan for their continued help and advice.

## Author contributions
A.T.M. and D.V.V. led the effort reported here; and J.G., X.L., J.P., C.S. and M.C.W. each contributed extensively and over several years to the effort. A.T.M. drafted the manuscript. The manuscript was reviewed, revised, and improved by all authors. All authors read and approved the final manuscript.

## Competing interests
The authors declare no competing interests.
