## [Peer Review File · Communications Biology]

Reviewers' comments:

Reviewer #1 (Remarks to the Author):

Precis

The authors have submitted a commentary about research culture and feedback from the Harvard Medical School (HMS) and associated institutions to the Journal for publication consideration. To address issues related to research culture, the authors have conducted several exercises to identify what's on the minds of members of HMS (and associated institutions). The authors report 'results', namely what they heard from the respondents. The commentary ends with a look to the future – what the authors/HMS would like to do.

Assessment The topic is important and getting global attention, some of which is referenced in the introduction section of the paper. I think the commentary is interesting but lacks specifics, particularly if someone from another institution wanted to reproduce this initiative in their environment. While I appreciate the paper is a commentary it needs some specifics to be interpreted. Maybe specifics are not part of a commentary ... but for a topic that discusses notions of reproducibility, I think some degree of detail is needed. What if a reader wants to carry out something similar at their institutions. Some details about the workshops/surveys; response rates/etc.

My comments below are more questions that I think need clarification in the paper:

1. The paper appears to be focused on laboratory – primarily talking about preclinical research – or is this broader? Laboratory is not a term commonly used when discussing clinical research teams/settings.
2. In an era when open science is starting to result in funder mandates (e.g., NIH) and early career researchers (ECRs)/faculty are promoting it, I was surprised open science was mentioned only once. In many research institutions (and increasingly funders) there are many requirements to ensure equity, diversity, and inclusiveness (and accessibility) – peripherally mentioned in the paper. Research integrity was mentioned twice. Is this because other terms have been used. I ask these questions, not to be a counter and assessor of word usage, but to reflect what I see in the literature and what I see at my institutions and institutions I visit. I'm really asking whether what the authors report is unique to Harvard, how the background work was developed? It does not seem to generalize to other organizations.
3. What I don't get, is any sense of the resource cost/time required to carry out the 'what was done' and more importantly, what it would take to do 'what we would like to do'. This is important because many organizations may well have similar aspirations as Harvard, they likely don't have the resources, immediately.
4. From reading the manuscript, I could not get a sense of the proportions of faculty involved/surveyed – senior; ECRs. The latter is important as they have been shown not to reflect the perspectives of more senior and often faculty with very traditional views. Similarly, men and women have been shown to differ in their perspectives in a variety of issues discussed here. Yet, I don't any sense whether there were differences in feedback to the surveys/etc. I think these issues need commenting on.
5. Is the finding that team practice (in the paper called laboratory culture) and mentorship new? Is there not a literature of these issues already?

6. Questionable research practices are also not a new concept in the context of what the authors learned. That said, the prevalence of the problem does not seem to have elevated itself to the paper. Similarly, the sheer depth of the problem is not there either (does this reflect the sampling used for the background work?). Does this reflect how the surveys/workshops/etc. were conducted and/or the respondents don't see these issues as problems? For example, during the COVID-19 pandemic we've had more retractions in a 2-year period than ever reported (see Retraction Watch). Authors have submitted papers in which the data never existed. Did these issues percolate to any discussion?

7. In the transparency section of the paper, there is no mention of the role of patient partners in research culture. Is this because patients/public (<https://doi.org/10.1016/j.ebiom.2021.103484>) are not involved in preclinical research at Harvard? I think the authors need to link this section to open science and/or the Transparency and Openness Promotion (TOP) initiative. While primarily focused on journals, is there a reason why universities can't take these TOP 'principles' onboard? Did the authors received any feedback on recommending the introduction of a chapter on reproducibility for all theses?

8. The last point I'll make, is whether the author's proposal moving forward reflects what the broader community is trying to do. For example, how does Harvard's proposal fit with that of CoARA, DORA, etc.? Is Harvard involved in the Helios project, and if so, did this work come out in the inhouse work or is it part of an initiative going forward?

I apologize for not having more time to add references to certain sections.

Reviewers' comments:

Reviewer #1 (Remarks to the Author):

Precis

The authors have submitted a commentary about research culture and feedback from the Harvard Medical School (HMS) and associated institutions to the Journal for publication consideration. To address issues related to research culture, the authors have conducted several exercises to identify what's on the minds of members of HMS (and associated institutions). The authors report 'results', namely what they heard from the respondents. The commentary ends with a look to the future – what the authors/HMS would like to do.

Assessment The topic is important and getting global attention, some of which is referenced in the introduction section of the paper. I think the commentary is interesting but lacks specifics, particularly if someone from another institution wanted to reproduce this initiative in their environment. While I appreciate the paper is a commentary it needs some specifics to be interpreted. Maybe specifics are not part of a commentary ... but for a topic that discusses notions of reproducibility, I think some degree of detail is needed. What if a reader wants to carry out something similar at their institutions. Some details about the workshops/surveys; response rates/etc.

In order to add specifics and greater detail to the manuscript, we have created two new figures. Figure 1 shows the steps we followed in conducting our R3 effort. Figure 2 gives an indication of the topics we addressed in our retreats and committee meetings. While the specific steps and topics discussed will vary depending on the institutional environment, these details may be useful to another institution that is interested in engaging in a similar R3 project. We referenced these figures on page 2 of the manuscript in the 'Our approach' section.

My comments below are more questions that I think need clarification in the paper:

1. The paper appears to be focused on laboratory – primarily talking about preclinical research – or is this broader? Laboratory is not a term commonly used when discussing clinical research teams/settings.

Our participants included basic research scientists as well as clinical and translational research scientists. We used the term 'laboratory' to encompass all of these research settings. We have indicated this on page 3 of the manuscript in the 'Laboratory culture' section.

2. In an era when open science is starting to result in funder mandates (e.g., NIH) and early career researchers (ECRs)/faculty are promoting it, I was surprised open science was mentioned only once. In many research institutions (and increasingly funders) there are many requirements to ensure equity, diversity, and inclusiveness (and accessibility) – peripherally mentioned in the paper. Research integrity was mentioned twice. Is this because other terms have been used. I ask these questions, not to be a counter and assessor of word usage, but to reflect what I see in the literature and what I see at my institutions and institutions I visit. I'm really asking whether what

the authors report is unique to Harvard, how the background work was developed? It does not seem to generalize to other organizations.

In the Introduction we refer to ‘transparency in science’ and have used this term throughout rather than ‘open science’ to encompass the range of terms that have been used, including, ‘open science’, ‘open access’ (referring primarily to publications), ‘reproducibility’, or as is the case in the OSTP memos ‘public access’ (reference 21). Each of these terms has slightly different nuances, but they all refer to efforts to ensure transparency in science. References 14-18 report on different aspects of transparency, including the National Academies reports *Open Science by Design* (reference 14) and *Reproducibility and Replicability in Science* (reference 15). Reference 41 provides a framework for rigorous and transparent research, and reference 42 requires ‘data sharing’, another important aspect of transparency in science. The discussions we have had with our colleagues are not unique to Harvard, and in our work with our faculty, we have emphasized the extensive work that is being done at the national and global level on these topics. (We note this in the last sentence of our Introduction.)

3. What I don’t get, is any sense of the resource cost/time required to carry out the ‘what was done’ and more importantly, what it would take to do ‘what we would like to do’. This is important because many organizations may well have similar aspirations as Harvard, they likely don’t have the resources, immediately.

The newly added Figure 1 indicates the time span for our project to date. After initial background work, the retreats and committee meetings spanned a couple of years, slowed down a bit by the COVID-19 pandemic. Figure 1 also indicates that the resources expended included some part-time staff work, supported by the Office of Academic and Research Integrity and the Office of Research Operations & Global programs, the funding of a full-time postdoctoral fellow through an internal grant, and volunteer work by all of the faculty who participated. We are currently in discussion about what resources would be needed to accomplish ‘what we would like to do’. We have added a sentence to that effect in the introduction to that section on page 5.

4. From reading the manuscript, I could not get a sense of the proportions of faculty involved/surveyed – senior; ECRs. The latter is important as they have been shown not to reflect the perspectives of more senior and often faculty with very traditional views. Similarly, men and women have been shown to differ in their perspectives in a variety of issues discussed here. Yet, I don’t any sense whether there were differences in feedback to the surveys/etc. I think these issues need commenting on.

Figure 1 notes that our faculty committees included both senior and junior faculty and both men and women faculty. We have added a statement about the proportions of these groups on page 3 where we discuss the faculty committees. We did not make note of significant differences in their perspectives and so did not comment on this in the manuscript.

5. Is the finding that team practice (in the paper called laboratory culture) and mentorship new? Is there not a literature of these issues already?

There is a literature on these issues, and we have referenced some of it (references 22-30). Our discussions were informed by that literature, and the discussions themselves were focused on what might be possible in our environment.

6. Questionable research practices are also not a new concept in the context of what the authors learned. That said, the prevalence of the problem does not seem to have elevated itself to the paper. Similarly, the sheer depth of the problem is not there either (does this reflect the sampling used for the background work?). Does this reflect how the surveys/workshops/etc. were conducted and/or the respondents don't see these issues as problems? For example, during the COVID-19 pandemic we've had more retractions in a 2-year period than ever reported (see Retraction Watch). Authors have submitted papers in which the data never existed. Did these issues percolate to any discussion?

We discussed questionable research practices at length and note this on page 4 in the section on 'Questionable research practices'. We have added several references (references 43-45) to the paper to make clear that this concept is not new, and that we are aware of the literature on this topic.

7. In the transparency section of the paper, there is no mention of the role of patient partners in research culture. Is this because patients/public (<https://doi.org/10.1016/j.ebiom.2021.103484>) are not involved in preclinical research at Harvard? I think the authors need to link this section to open science and/or the Transparency and Openness Promotion (TOP) initiative. While primarily focused on journals, is there a reason why universities can't take these TOP 'principles' onboard? Did the authors received any feedback on recommending the introduction of a chapter on reproducibility for all theses?

We did not discuss the role of patient partners in research in this project. We also did not directly discuss the thesis requirements for our PhD students. These are clearly laid out in the student resources/handbook and include a section on making the thesis openly available after an allowable embargo time (See <https://dms.hms.harvard.edu/dissertation-and-defense>), but they do not currently include the recommendation for a chapter on reproducibility. We are in the process of designing a curricular framework (noted on page 6 of the manuscript) for our PhD students, and we will consider the reviewer's comments.

8. The last point I'll make, is whether the author's proposal moving forward reflects what the broader community is trying to do. For example, how does Harvard's proposal fit with that of CoARA, DORA, etc.? Is Harvard involved in the Helios project, and if so, did this work come out in the inhouse work or is it part of an initiative going forward?

Harvard University is a member of the Higher Education Leadership Initiative for Open Scholarship (HELIOS Open). We are aware of the San Francisco Declaration on Research Assessment (DORA), but we were not aware of the recent CoARA agreement and principles. We thank the reviewer for bringing these to our attention. The Hong Kong Principles (reference 19) support and make reference to DORA, and likewise both CoARA and the Hong Kong Principles

support DORA as well as the Leiden Manifesto (reference 20). All of these efforts will help inform our work on improved methods for researcher assessment and academic advancement.

I apologize for not having more time to add references to certain sections.

We thank the reviewer for these thoughtful comments. They have helped improve our commentary.

REVIEWERS' COMMENTS:

Reviewer #1 (Remarks to the Author):

I have reviewed the revised version of the manuscript and the authors responses to my comments. I think the two new figures add to the understanding of the paper.

I'm satisfied with the authors responses to my previous review.

I have not additional comments.